# A scalable culturing system for the marine annelid *Platynereis dumerilii*

Emily Kuehn[1], Alexander W. Stockinger[2], Jerome Girard[1], Florian Raible[2], B. Duygu Özpolat[1] *

**1** Marine Biological Laboratory, Woods Hole, Massachusetts, United States of America, **2** Max Perutz Labs, University of Vienna, Vienna, Austria

* dozpolat@mbl.edu

**Data Availability Statement:** All relevant data are within the manuscript and its Supporting Information files.

**Funding:** BDO received funding from Hibbitt Startup Funds. FR received funding from projects P30035, and I2972, by Austrian Science Fund

## Abstract

*Platynereis dumerilii* is a marine segmented worm (annelid) with externally fertilized embryos and it can be cultured for the full life cycle in the laboratory. The accessibility of embryos and larvae combined with the breadth of the established molecular and functional techniques has made *P. dumerilii* an attractive model for studying development, cell line-ages, cell type evolution, reproduction, regeneration, the nervous system, and behavior. Traditionally, these worms have been kept in rooms dedicated for their culture. This allows for the regulation of temperature and light cycles, which is critical to synchronizing sexual maturation. However, regulating the conditions of a whole room has limitations, especially if experiments require being able to change culturing conditions. Here we present scalable and flexible culture methods that provide ability to control the environmental conditions, and have a multi-purpose culture space. We provide a closed setup shelving design with proper light conditions necessary for *P. dumerilii* to mature. We also implemented a standardized method of feeding *P. dumerilii* cultures with powdered spirulina which relieves the ambiguity associated with using frozen spinach, and helps standardize nutrition conditions across experiments and across different labs. By using these methods, we were able to raise mature *P. dumerilii*, capable of spawning and producing viable embryos for experimentation and replenishing culture populations. These methods will allow for the further accessibility of *P. dumerilii* as a model system, and they can be adapted for other aquatic organisms.

## Introduction

Nereidid worms such as *Platynereis* have been popular in studies of development and fertilization because of transparent, abundant, and comparatively large eggs and embryos [1,2]. As researchers like Edmund Beecher Wilson did in the late 19th century, many labs today benefit from *Platynereis* as a model organism for addressing a wide range of biological questions such as cell type evolution, nervous system evo-devo and activity, reproductive periodicity, circalu-nar cycling, endocrinology, regeneration, post-embryonic segment addition, stem cell biology, fertilization, oocyte maturation, embryonic and larval development [3–16]. The sexual worms broadcast spawn, producing thousands of externally-developing embryos. The embryos are

(FWF). The funders had no role in study design, data collection and analysis, decision to publish, or preparation of the manuscript.

**Competing interests:** The authors have declared that no competing interests exist.

large enough to inject but small (about 160 μm in diameter) and transparent enough to image live and fixed samples. *Platynereis* has a relatively quick and highly synchronized embryogenesis: it takes only 18 hours from fertilization to hatching as a planktonic trochophore larva in *P. dumerilii* [2,17]. This allows researchers to study embryonic development over the course of just one day. A number of tools and techniques are already established in *P. dumerilii* [3] including microinjection [5,12,18], transgenesis and genetic tools [18–23], single cell RNA sequencing [24], behavioral tracking [8,16,25,26], and live imaging [5,27]. This well-equipped tool kit combined with the large number of embryos generated by each fertilization make *P. dumerilii* an attractive model organism which can be cultured under specific laboratory conditions for its full life cycle.

Ernest E. Just was among one of the earliest of people who tried rearing *Platynereis* in the lab at the beginning of the 20[th] century [28]. He studied fertilization in *Platynereis megalops*, which he collected from Great Harbor in Woods Hole [29], and wanted to have access to eggs throughout the year instead of only during the summer. In Europe, studies of *P. dumerilii* go back to early 20[th] century [30] at Naples Zoological Station. *P. dumerilii* cultures today are thought to mostly originate from the Bay of Naples, and have been bred in the lab since 1950s (originally by Carl Hauenschild) [2,17]. Even though resources for culturing *P. dumerilii* exist including Fischer and Dorresteijn's excellent guide online [31], these resources only provide guidelines for larger scale culturing of *Platynereis* and detailed guidelines for establishing small (but scalable) culturing are not available. For many research areas (such as physiology, behavior, aging, reproduction. . .) there is also the need for flexibly adjusting the environmental parameters, which is challenging with the traditional methods of culturing *P. dumerilii*. Finally, several areas for standardizing culturing methods remain to be established, especially regarding the feeding methods. This is particularly important for studying biological processes that are affected dramatically by nutrition, and for being able to carry out experiments comparable across different research labs.

Here we describe a scalable, small footprint setup for culturing *P. dumerilii*, including detailed methods of light regulation, light and temperature monitoring, husbandry, and feeding. This setup, with blackout curtains and its own automatic lighting which serves as the sun and moon, removes reliance on dedicated culture rooms. It provides greater flexibility for choosing and adjusting culturing components (such as light source) and can be put together at a lower cost than similar designs that use incubators in place of a shelving unit [32]. We provide worm maturation data that can be used to scale the culture up or down, based on the number of mature worms needed. We also present standardized feeding methods we developed using powdered, commercially available nutrients such as spirulina. This homogenous suspension can be distributed throughout the culture boxes evenly. Our feeding method eliminates the ambiguity associated with using fresh or frozen spinach, and will enable better comparison of data across laboratories especially for types of research, such as physiology, where diet is a particularly important factor. Finally, we present data and images of *P. dumerilii* embryos, larvae, juveniles, and adults we obtained from our cultures, and visualization of normal development, confirming the robustness of the culturing conditions.

## Materials and methods

### Water types and filtering

*P. dumerilii* embryos, larvae, juveniles and adults were kept in full strength natural filtered sea water (NFSW). The sea water was first filtered through a 1 μm filter system in the Marine Resources Center at the Marine Biological Laboratory (MBL). NFSW was used primarily for juvenile and adult culture boxes (S1A Fig). The pH of the NFSW used at the MBL typically fell

within the range of 7–7.5, but has been observed to fluctuate seasonally. The salinity of the MBL's NFSW is 32–34 parts per thousand (ppT). Some of the experiments were carried out in Vienna at the Max Perutz Labs, at comparable salinity levels (34–35 ppT).

For food preparation, microinjections, and culturing embryos and young larvae, sea water was filtered further via a glass graduated filtration funnel (*Pyrex*, 33971-1L) secured to a glass funnel stem (*Sigma*, Z290688) by a spring clamp (*Millipore*, xx1004703) (S1B Fig). On top of the funnel stem, a 0.22 μm pore size nitrocellulose membrane (*Millipore*, GSTF04700) was placed, and this membrane was covered by one piece of Whatman filter paper (*GE Healthcare*, 1002055). The filter unit was placed on a glass vacuum flask (*Fisherbrand*, FB-300-4000) fitted with a rubber adapter (*Fisherbrand*, 05-888-107) so that the stem sat snugly in place (S1B Fig).

## Collecting embryos for sustaining cultures

To spawn mature worms, male and female worms were placed together in a small glass dish of approximately 150 mL 0.22 μm NFSW. After spawning, worms were removed from the dish, sea water with excess sperm in the dish was discarded and replaced with clean sea water to prevent polyspermy, and embryos were kept at 18°C. Around 24 hours post fertilization (hpf), larvae were poured onto a 85 μm sieve and 0.22 μm NFSW was poured over the embryos several times to wash the jelly off. To make a 85 μm sieve, we cut off the conical end of a 50 mL tube (and removed the cap), stretched a 85 μm pore mesh (*Component Supply*, U-CMN-85-A) over one of the ends of the tube, and secured the mesh with a rubber band or glue.

After the jelly was washed off, larvae were pipetted off the sieve and into a clean dish of 0.22 μm NFSW, or the sieve was flipped over an empty dish and larvae were washed into the dish. Penicillin Streptomycin (*Gibco*, 15140–122) (1:1000) was added, and the cultures were kept at 18°C until 7–10 days post fertilization (dpf).

## Setting up culture boxes—High and low density populations

As containers for culturing, Sterilite brand boxes made of polypropylene and polyethylene (PP5 type non-reactive plastic) were used (Fig 1G). These boxes are non-toxic, inexpensive, come in several sizes, and have lids that can sit loosely to allow air in and out when the latches are not used. To ensure no residual chemicals or dust from production remained, the boxes were rinsed with deionized water, soaked overnight, and washed with a clean sponge without any detergent before they were used for cultures for the first time.

Healthy larvae at 7–10 dpf were transferred into Large Sterilite boxes (35.6 cm x 27.9 cm x 8.3 cm) (*Sterilite*, 1963) in 1.5 L 0.22 μm NFSW (no air bubbler needed at this stage), and were kept in the 18°C incubator until around 1–2 month(s) post-fertilization (around the time young juvenile worms started building tubes). At this time, these cultures were transferred from the incubator to the shelving unit to begin assimilation to the lighting schedule. These large boxes contained a high-density population of worms of about 300–350 per box. To facilitate the dispersal of oxygen throughout these cultures with growing worms, high density boxes were equipped with air bubblers (*Tetra*, Whisper Air Pump, 77846) once transferred to the shelf. Bubbling was done at maximum valve opening of the pumps, and this provided a gentle bubbling rate in the middle of the culture box. Air tubes were inserted by making a hole on the lid using a hot glue gun tip (using the heat to melt the plastic, without the glue).

Low density cultures were established a little over two months (typically at the beginning of the third month) after the worms were born. Using a paint brush, the worms were pushed gently to come out of the tubes and were collected with a pipette to be transferred to a low density box. These cultures consisted of 30 worms per small Sterilite box (27.9 cm x 16.8 cm x 7 cm) (*Sterilite*, 1961) in 500 mL 1-μm NFSW, and were not aerated.

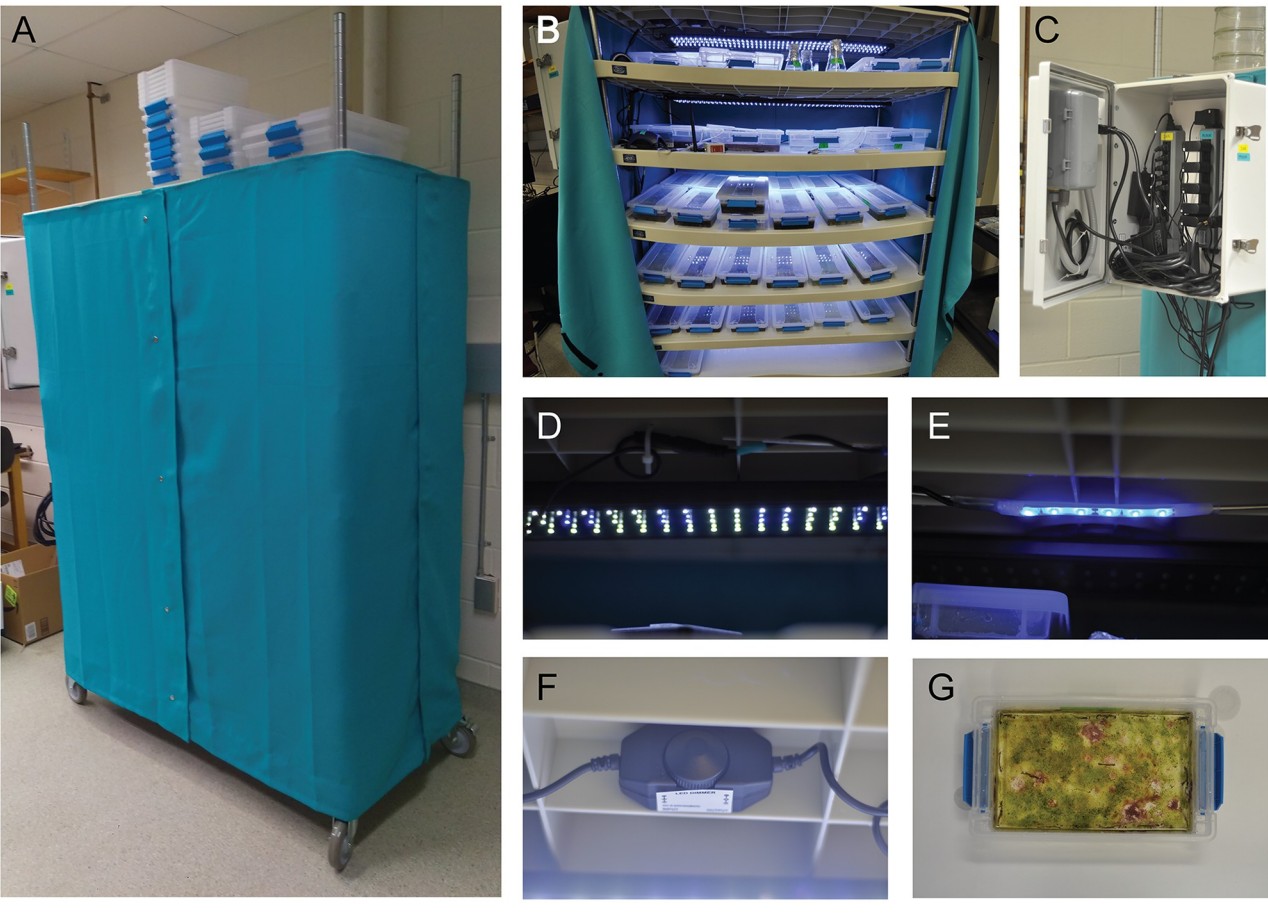

**Fig 1. Scalable culture setup for *Platynereis dumerilii*.** A) The shelving unit is enclosed in a blackout curtain which keeps unwanted light out during "moon off" periods of *P. dumerilii*'s lunar maturation cycle. Putting the unit on wheels (arrowhead) allows us to easily access culture boxes at the back of the shelves. The curtain can also be opened from both sides. B) Each shelf within the unit is equipped with identical lighting to ensure all culture boxes are receiving both sun and moonlight. C) The lighting is controlled automatically. Controls are located in a box outside of the unit. D) Sun lighting on. E) Moon lighting on. F) The intensity of moonlight can be easily adjusted with the dimmer shown here. A wireless light monitor was used to ensure each shelf received equal light intensity. G) Sterilite boxes made of polypropylene and polyethylene were used to house worms. After several months of culturing, algae was seen to grow in the boxes without harmful effects to the worms.

### Feeding *Platynereis* larvae

Depending on availability at our facilities, we used either *T-iso*, or *T. chuii*, or a mix of both algae species to feed the young *P. dumerilii* larvae (starting around 7–10 dpf). The algae cultures were kept in a room with ample natural light and/or with additional LED lights. Algae cultures were grown in glass or plastic containers and were sometimes aerated to ensure faster growth. As culture medium, a 50X stock of Guillard's F/2 marine water enrichment solution (*Sigma*, G0154-500ML) was diluted in sterilized 0.22 μm NFSW [33]. NFSW was sterilized by heating to 80˚C for 6 hours. Algae were collected into 50 mL centrifuge tubes when cultures were seen to be dense enough (dark brown or green color, depending on the species). The tubes were then centrifuged for 10 minutes at 2000 rpm. The supernatant was discarded, and the tube was refilled with additional algae stock. Centrifugation was repeated until the pellet is large enough to fill the conical part of the tube ($\sim 2.5$mL). Then algae pellet was resuspended in 50 mL of 0.22 μm NFSW. The density of algae feed prepared this way was determined using a hemacytometer by counting cells, and ranged between $3^*10^6$ and $6^*10^6$ cells/mL. These algae

stocks could then be stored at 4˚C for several weeks for later use. During the first month of development, 25 mL per large box (1.5 L water volume) of this concentrated stock was distributed using a transfer pipette, twice per week. On the second month of development, the juvenile worms were switched to the spirulina regimen (see Results and discussion below for details).

## Maintaining the larval and juvenile culture boxes

**Maintaining larvae.** The larvae may perish easily if water cleanliness is not maintained. To ensure optimal water cleanliness larval culture boxes were checked under the microscope at least once a week. If growth of protozoans was observed, water was removed using a sink-enabled vacuum through a 85-µm sieve which is large enough to let protozoa out but prevents removing small larvae. 0.22µm NFSW was used to culture the larvae.

**Maintaining juveniles.** The NFSW in the culture boxes was completely replaced every two weeks with new 1.0 µm NFSW. For boxes with juvenile worms that have already formed tubes, dirty water was poured completely into a plastic dishpan, as the worms mostly stay in tubes. Any escaped worms were transferred back into the culture box from dishpan. The dirty water was discarded into a container to which bleach was added prior to final disposal into the drain, in order to prevent introducing *P. dumerilii* into the environment. For cultures with very young juveniles, dirty water was removed using a vacuum filter (as with larval cultures above).

## Mature worm collection and maintenance

Worms were determined to be sexually mature once the dramatic change in body color (red for males, yellow for females) took place or when the worms were observed to be swimming at the surface of cultures, a behavior uncharacteristic of juvenile worms. Mature worms were collected and separated into females and males in large Sterilite boxes equipped with air bubblers. Mature worms for which the sex could not yet be identified at the time of collection ("unknown sex" worms) were placed with the males, as males are thought to not spawn in the presence of immature females. The "unknown sex" worms displayed the color change and loss of gut content, but did not appear fully yellow or red. The number of worms for each category ("female", "male", "unknown sex") was recorded at each time of collection and graphs were plotted based on these numbers. The boxes were monitored daily to remove dead worms, change the water if needed, and remove and use mature worms for setting up new cultures and for experiments. In order to reduce maintenance effort, mature worm collection was restricted to only 2 weeks (on Mondays, Wednesdays, and Fridays) of a given month when the worm maturation was expected to peak (starting about 10 days following the last day of "moon on").

## Temperature control and monitoring

Most labs use 18˚C as the culturing temperature for *P. dumerilii* for the full life cycle. Previous studies on embryonic and larval development suggest that at least for these early stages the animals can be kept at lower or higher temperatures (14–30˚C) [17], while a systematic testing of temperatures higher than 18˚C for the full life cycle has not been reported. We have found that our cultures tolerated slightly higher average temperatures (19–20.5˚C) (S2 Fig). The thermostat of the culture room was set to 20˚C. However, we kept an additional portable AC unit (*Arctic King*, WPPH08CR8N) set to about 18˚C (65˚F) running next to the cultures, which served as a back-up. We also used a large 18˚C incubator (no light cycles) for keeping some worm cultures as reserves, in case the room temperature control severely failed.

To monitor temperature, a Monnit wireless temperature monitor (*Monnit*, MNS-9-IN-TS-ST) was placed on one of the shelves. In addition, we also used a thermometer (*Suplong*, COMINHKPR144821) which stores the minimum and maximum temperatures recorded within the shelving unit for manually checking the temperature fluctuations. The data collected by the Monnit thermometer was stored online and provided a complete list of all temperature readings. This system also sends alerts if it detects temperatures outside of a specified range.

### Light spectra measurements

For the sun (*Nicrew*, ZJL-40A) and moon (Ebay seller: *21ledusa*, 700381560185) light sources, the irradiance per wavelength was measured using a spectrometer (*International Light Technologies*, ILT-ILT950-UV-NIR). To achieve different light intensities, an LED dimmer (DC12V~24V, *Supernight*) was used. The spectrometer was set up at 20 cm distance to the light source in an otherwise dark room. For plotting the data, the sensor output (in $\mu W/cm^2$) was converted to the more commonly used photon flux (photons/$(m^{2*}s)$).

## Results and discussion

### Building the scalable shelving unit

The first consideration for a lab to begin culturing *P. dumerilii* is the availability of appropriate housing infrastructure. In nature, synchronized reproduction occurs in phase with the lunar cycle. To mimic this, *P. dumerilii* is typically maintained in a defined light regime, consisting of 16: 8 hours of daylight: night, with dim nocturnal lighting simulating a full moon stimulus for several nights within a month (see below). Most labs currently working with *P. dumerilii* keep their worms in a separate culture room and control the room lights to achieve day, night, and moon conditions. But having a dedicated worm culture room may not always be feasible or even needed if only a small scale of *P. dumerilii* culturing is desired. Additionally, *P. dumerilii* maturation peaks for only two weeks out of the month due to the lunar maturation cycle. Labs wanting to have mature worms continuously would need separate culture spaces on opposite lighting schedules.

To circumvent the need for separate culture rooms, and to use the space available for multiple purposes, we designed a stand-alone culture setup (Fig 1) (see S1 File for a comprehensive list of all the parts, reagents, and ordering information). The setup is composed of a shelving unit (*Nexel*, 188127) with wheels (*Nexel*, CA5SB) for ease of access, black-out curtains to prevent unwanted light reaching the worms during the night hours when complete darkness is needed (i.e. no "moon"), and two sets of lights (sun and moon) on separate circuits installed on each shelf controlled by a timer (Fig 1C, S2 File). With this setup, labs wanting to have mature worms available during the entire month could have two or more culture units on opposite moonlight schedules in the same room.

The shelving unit (Fig 1A and 1B) was assembled according to the manufacturer's directions. The unit comes with 4 shelves. We added 3 more shelves (*Nexel*, S2448SP), each being approximately 8 inches (20.3 cm) apart from each other. This allowed enough space for the lights and stacking up 2 rows of culture boxes. Small (low worm density) boxes can fit into two rows on each shelf of our culturing unit. They can also be stacked, thus in theory 28 small Sterilite boxes (14 if unstacked) can be stored per shelf.

After assembling the shelving unit, the sun (*NICREW*, B06XYKD67V) (Fig 1D) and moon (Ebay seller: *21ledusa*, 700381560185) (Fig 1E) lights were installed on each shelf by punching holes on the plastic and using aluminum welding rods to secure the lights (Figs 2E, 2E' and 3B). See Figs 2 and 3 for circuit diagrams and blueprints for the assembly of lights. Dimmers

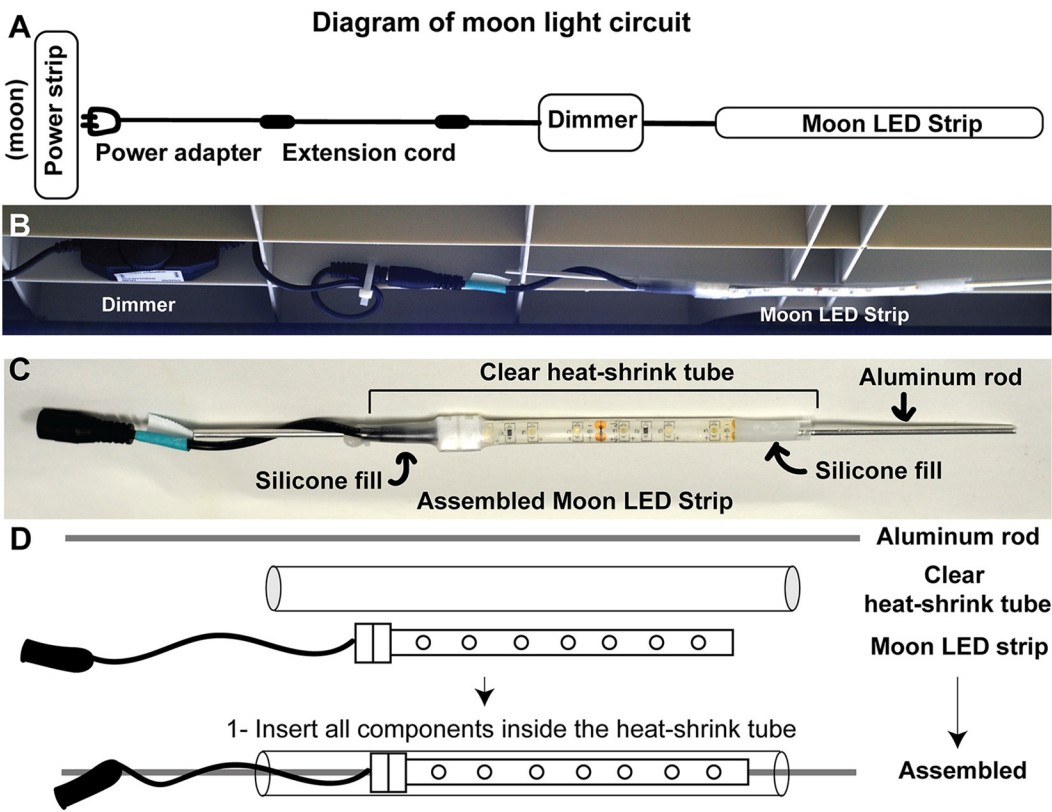

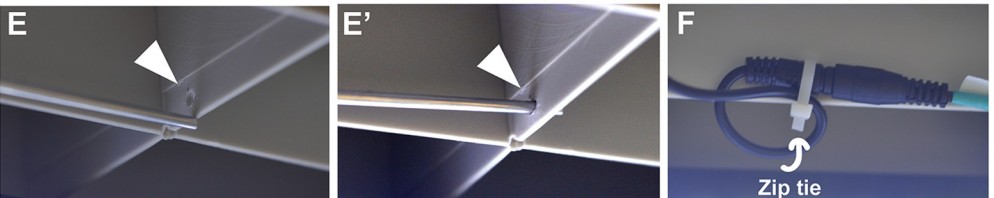

**Fig 2. Setting up the moon light.** A) Diagram of moon light circuit. One moon light per shelf is installed, connected to a dimmer and an extension cord (for reaching to the power strip). The moon power adapters should be plugged into the power strip controlled by the timer channel dedicated to the moon lights. (Note that power adapters come with the LED strips and do not need to be purchased separately). B) Photograph showing the dimmer and LED strip components, attached to the shelf. C) Components of the moon light LED strip. The LED strips are purchased and then modified for the shelving. Aluminum rod helps with attaching the light to the shelf. Clear heat shrink tube and silicone is for sealing the light for protection from water. D) Schematic showing the components of the moon LED light, and assembly instructions. E) Using a hole puncher, plastic shelving is modified for insertion of the aluminum rod. Arrowheads: holes F) Holes can also be used for securing cables with zip-ties.

(*HitLights*, B00RBXPDQU) were included in the circuits to control the brightness from each set of lights (Figs 1F and 2B).

Next, 4 pieces of blackout curtains (*Deconovo*, B01MU1CMSD, 42x63 inches) were modified by sewing hook and loop (e.g. Velcro) strips on the curtains along the edges (Fig 4). By using Velcro strips we could also ensure that the curtain was reliably secured in the same position after each time the cultures were accessed during daily operations. This is also made possible by the snug fit of the curtain around the shelf. We found that using 4 pieces of fabric

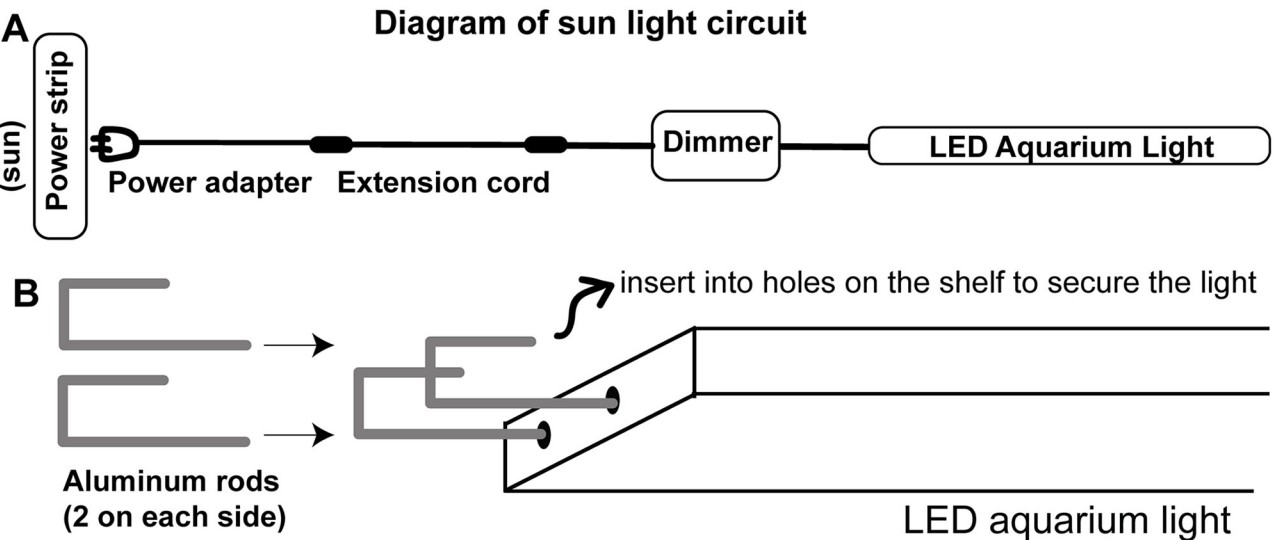

**Fig 3. Setting up the sun light.** A) Diagram of the sun light circuit. One aquarium light per shelf is installed to provide light during the day. The aquarium light is connected to a dimmer and an extension cord (for reaching to the power strip). The power adapters should be plugged into the power strip controlled by the timer channel dedicated to the sun lights. In addition, aquarium light switch should always be ON, the turn ON-OFF will be controlled automatically by the timer. B) Bent aluminum rods (2 on each side) are used for securing the light onto the shelf. See Fig 2E and 2E' for aluminum rod insertion and hole punching.

provided flexibility of access to cultures from all sides of the shelving unit, as well as ease of running cables from the lights and air pumps to the power box (Fig 4B). A basic sewing machine (Singer Simple 3232) and needles made for thick fabric were used for sewing the hook and loop strips onto the curtains.

After the blackout curtains were installed, the power box (*Allied Moulded Products Inc.* AM2068RT) was attached to one of the shorter sides of the shelving (either left or right side would be fine, depending on which side is more comfortable for a given space). The power supply box was attached to the shelves using two plastic struts (on the left and the right of the back panel) sandwiched between the box and shelves; the struts were helpful to support the screws securing the box onto the shelf. Next, power strips and the timer were installed into the power box (Fig 1C). The sun and moon lights were connected to the timer in separate circuits with the aid of an electrician.

## Light/dark cycles

*P. dumerilii* mature according to a lunar cycle. In their natural habitat, worms will congregate at the surface of the water within about ten days following the full moon for a period of about 2 weeks to spawn [30,34]. In the lab, a synchronized swarming pattern can be obtained by simulating 28-day lunar cycle lighting conditions [16]. This is achieved by having a period of 8 nights where a dim light turns on at night in order to mimic the moon. This period is followed by 20 days in which the worms are kept in complete darkness during the night hours. Note that the "moon" lights do not mimic the phases of the moon; worms respond to simple presence/absence of light at night. Setting the lunar lighting to a schedule of four exact weeks simplifies planning in terms of knowing when the worms will peak in maturation. We followed these standards established by others [31] in our culture conditions: For all days, worm cultures in the shelving unit received 16 hours of daylight and 8 hours of night (either with the moonlight on or off) (Fig 5A). These on and off cycles were controlled automatically by an

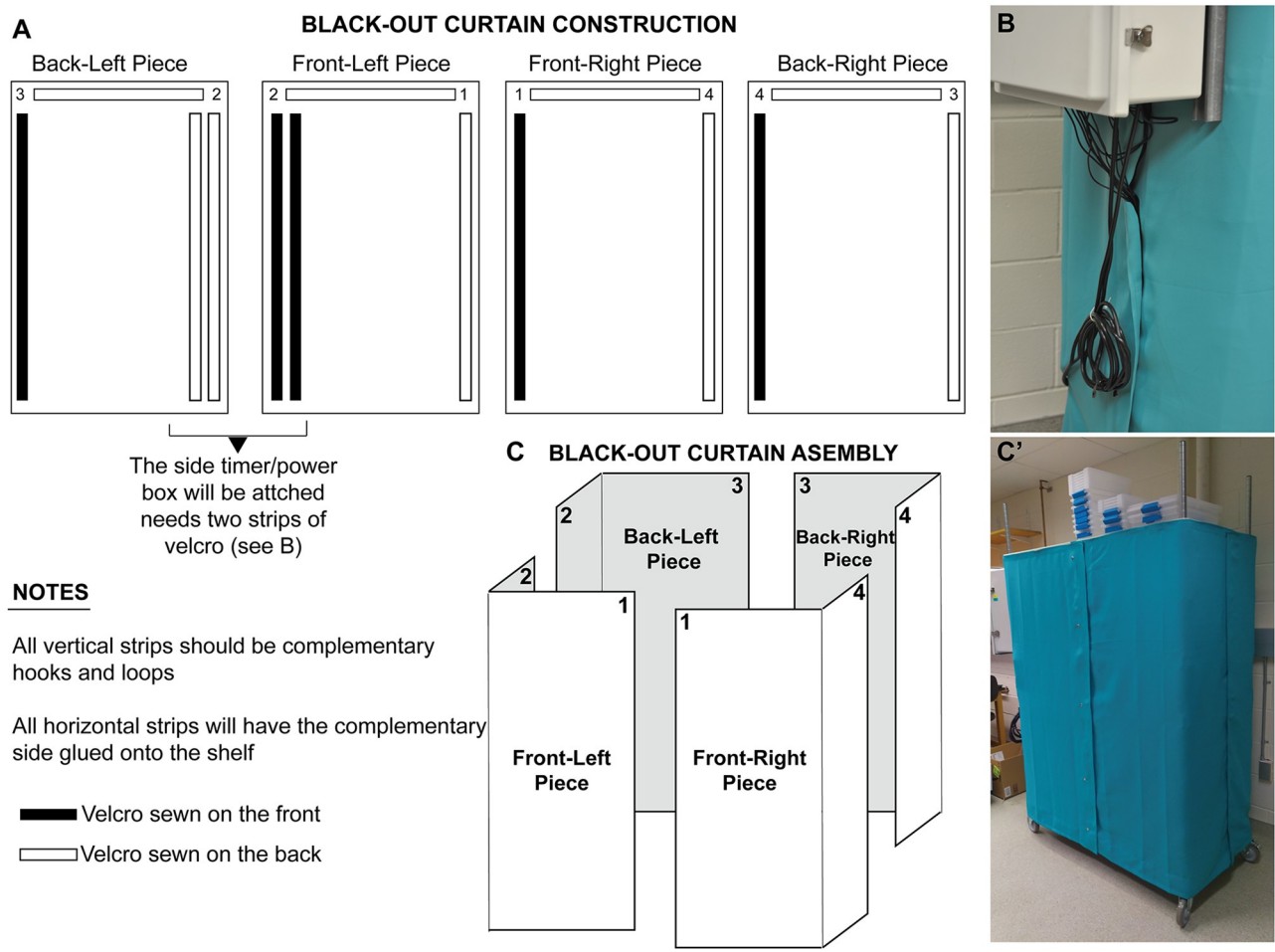

**Fig 4. Blackout curtain construction and assembly.** A) Instructions for sewing Velcro (hook and loop) strips onto the blackout curtains. Top parts of the curtains will have one side of adhesive Velcro sewn on, and the other side applied onto the shelf. Even if adhesive type of Velcro is used, the adhesive alone is not sufficient to keep Velcro strips on fabric, and these strips need to be sewn. Our light monitoring data show that the light signal recorded from the bottom shelf during "moon off" periods was the same as values recorded for the other shelves, indicating that the absence of Velcro along the bottom of the curtain was not a source of light contamination. B) The side of shelving that will have the power box installed will have two Velcro strips to allow for extra sealing for cables from the shelf to run outside to the box. C-C') Schematic showing the assembly of 4 curtain pieces to cover the shelving unit (C) and a picture of the assembled shelf (C'). Note that the uppermost shelf is used for storing culture boxes, and does not function for culturing worms as it is not covered with curtains.

industrial timer (*MRO Supply*, DG280A), and settings were adjusted according to manufacturer's user guide (see S2 File for timer settings). The timer removed the need for manually installing and uninstalling the moon lights.

To monitor that light conditions were properly set and maintained during the day and night, we used a Monnit wireless light monitor (*Monnit*, MNS-9-W2-LS-LM). The monitor sends the data to a wireless receiver installed at our facility, and users can access the sensor reads online via a simple user interface (https://www.imonnit.com/). The light monitor was periodically moved between shelves to ensure that the light signal reaching the worms was the same throughout the unit and that all the sun and moon lights were functioning properly. We then continued to regularly monitor each shelf lighting this way for light contamination during night hours of "moon off" periods or any lights that may need replacement.

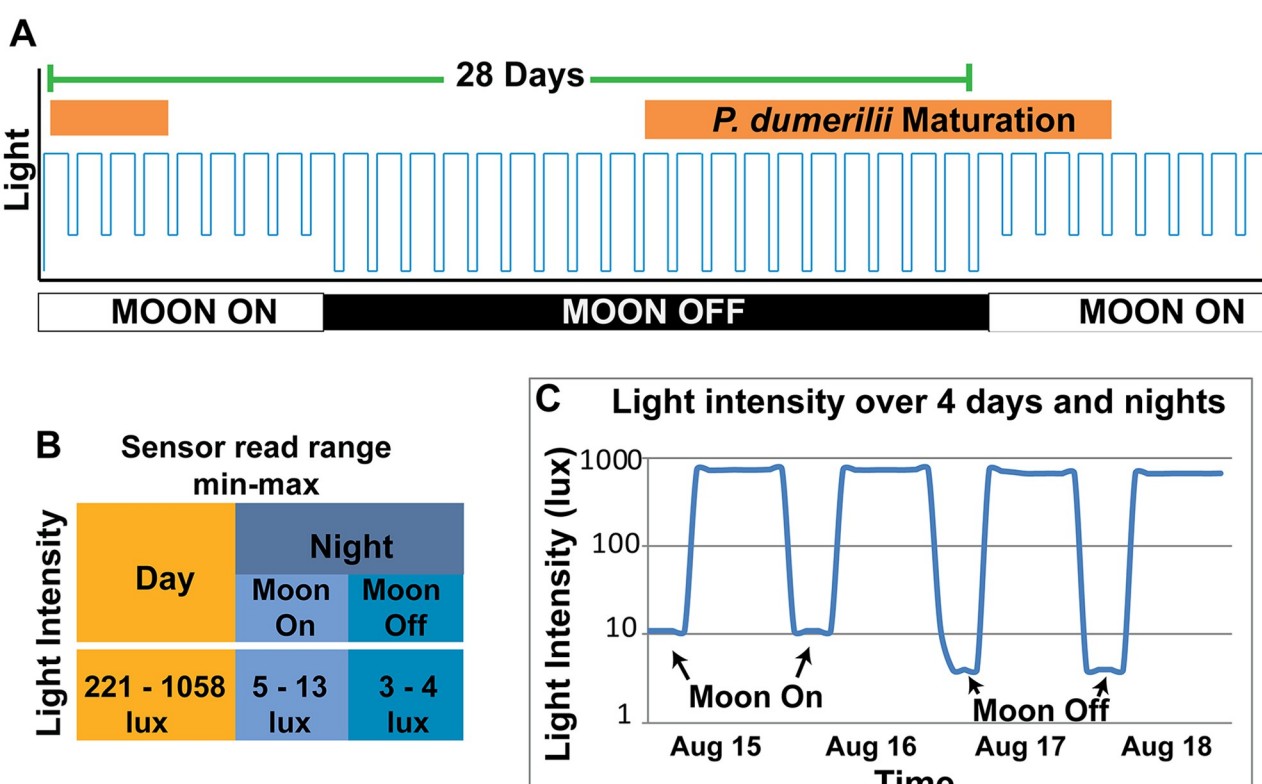

**Fig 5. Lighting schedule.** A) The schematic shows the lighting schedule we used to culture *P. dumerilii* through maturity. Worms received 16 hours of sunlight every day and 8 hours of either moonlight or complete darkness at night. In a 28 day cycle, the moon is on for 8 nights and off for the remaining 20 days. *P. dumerilii's* maturation period (indicated with the orange bar) begins 10 days after the last night the moon was on and lasts for 2 weeks. B) Values of light intensity (lux) were collected by a wireless light sensor. The sensor was placed inside a culture box in order to record the amount of light the worms were receiving through the translucent culture box lids. C) Example light sensor data recording over 4 days showing the transition between a period when the moonlight was on and off.

Using the light monitor, we collected several types of light intensity data. To measure the amount of light each shelf received, we rotated the light sensor between shelves recording light levels for 24 hours each rotation (Fig 5B and 5C). This was done for a period of two months so that we could have several readings from each shelf during "moon on" and "moon off" periods. For the first month, the sensor was placed on top of a Sterilite box. For the following month, the monitor was placed inside an unused small Sterilite box to get a more accurate estimate of the amount of light the worms received through the translucent lids. Of the measurements taken from inside a Sterilite box, the daylight range was between 221–1058 lux (see below for detailed photon flux information). Even though we did not systematically test whether the lowest or highest settings of illumination had different effects, we obtained mature worms from all shelves. Thus we conclude that values within this range will be sufficient for the worms to grow and mature.

The moonlight range measured from inside a Sterilite box was 5–13 lux during a "moon on" period and 3–4 lux during a "moon off" period (Fig 5B). To determine whether the reading of 3–4 lux during a moon-off period was due to sensor error or an outside light which infiltrated the unit's curtains, we placed the sensor in a completely sealed and opaque box, and the sensor still read 3–4 lux. Therefore, we assume that when the sensor read 3–4 lux, there was no or negligible light within the culture setup. Taking these values as "zero" for this particular

sensor, we adjusted the moonlight brightness to a higher range of brightness using the dimmers aiming for 10–15 lux (Fig 5C), taking previously-published values into account [16]. We placed the sensor in an empty box and placed a culture box on top to test if moonlight penetrated stacked cultures. We found that moonlight levels were lower (typically around 5 lux), but slightly above our "zero" value (3–4 lux). In this study, we did not systematically keep track of how, or if, the reduced moonlight affected worm maturation of stacked cultures, however we did not notice any differences in mature worm production by these boxes.

## Light spectra distribution of the moon and sun lights

The distribution and strength (irradiance) of the wavelength available to organisms can affect many biological processes such as circadian rhythm and sexual maturation [35,36]. We next determined the spectral distribution of the moon and sun lights in our culture setup by using a photo spectrometer. We tested the spectral properties of both light sources at different light intensities and measured the energy flux for each wavelength. We found that the moon lamp consistently emitted light around 450 nm wavelength at different intensity settings (one example shown in Fig 6A). We also found that the sun lamp emission spectrum did not change at two different levels of illuminance tested: 200 lux (Fig 6C) and 1000 lux (Fig 6D). Overall, these results indicate that these light sources can be used at any intensity needed for a given experiment without causing a change in the emission spectrum.

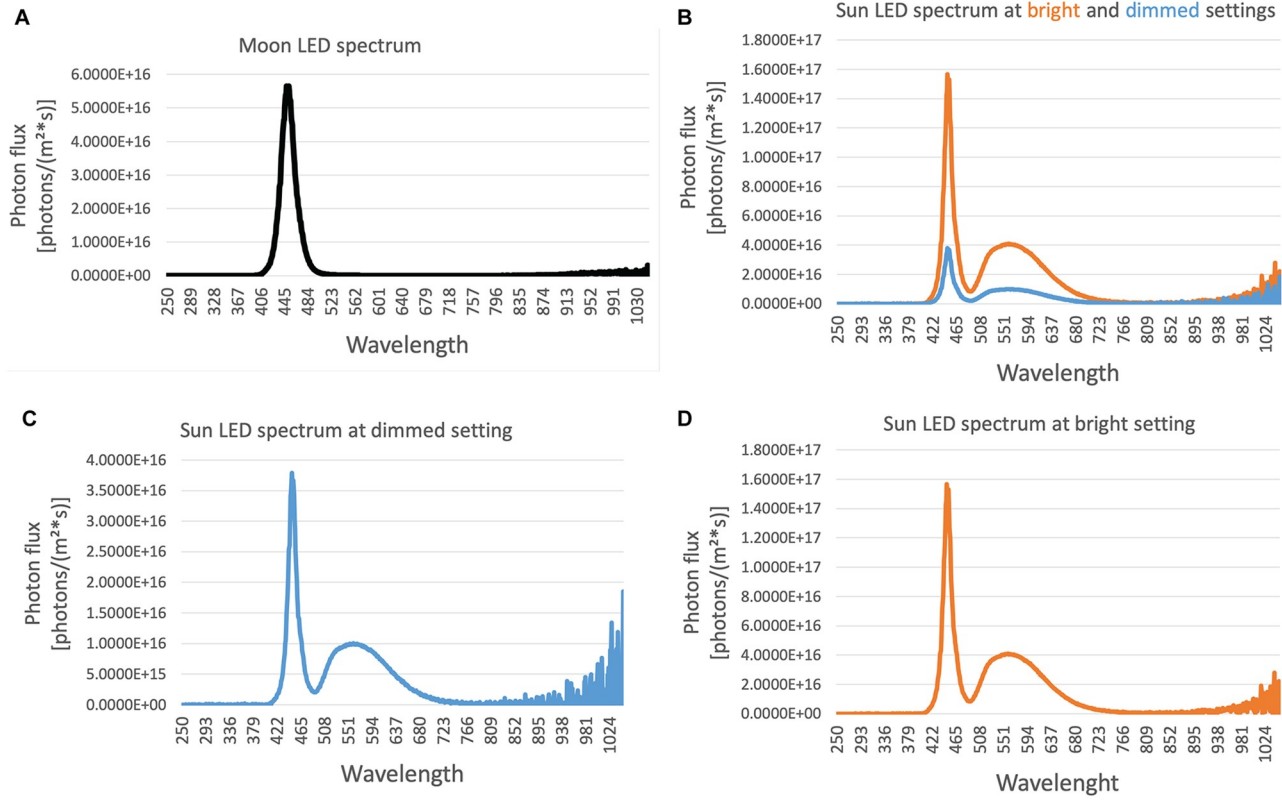

**Fig 6. The distribution of irradiance per wavelength of the moon and sun LEDs.** A) An example of moon LED light irradiance per wavelength is shown. The emission was around 450 nm at the different light intensities tested. B) Sun LED light spectral distribution did not change when the brightness of the lamp was changed between ~1000 lux (bright) and ~200 lux (dimmed). Graphs in (C) and (D) show the dimmed and bright setting distributions in (B) separately.

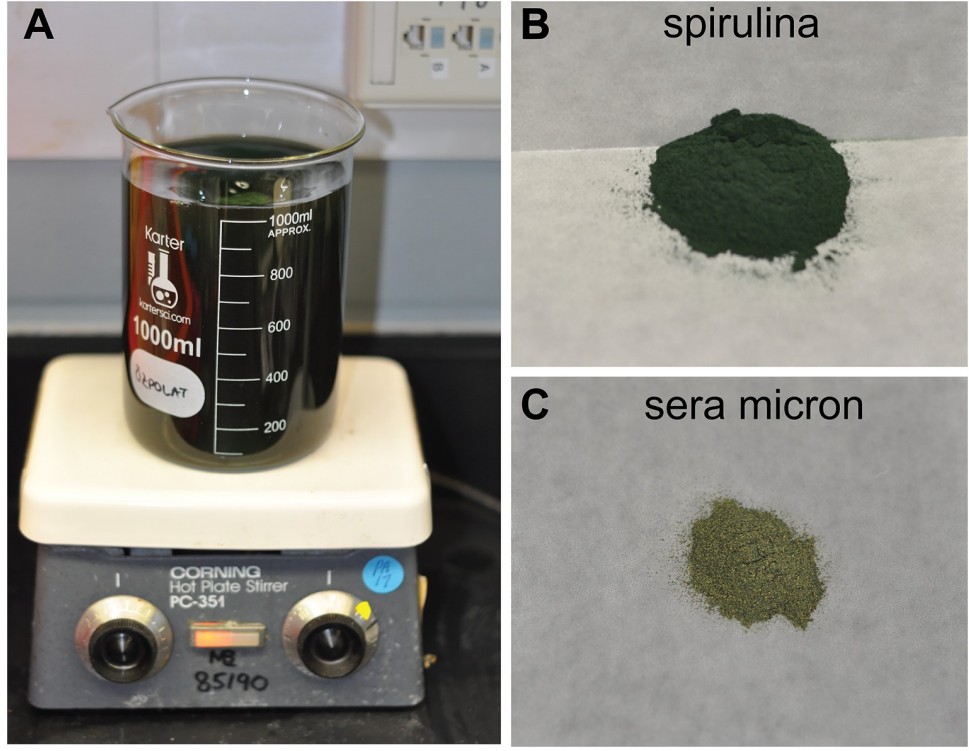

**Fig 7. Standardized feeding methods.** A) 1 g spirulina and 0.3 g sera micron flakes were added to 1 L NFSW and mixed thoroughly to create a homogenous solution. B) Spirulina powder. C) Sera micron growth food.

### Developing a standardized feeding method for juvenile *Platynereis dumerilii*

To date, most labs that culture *P. dumerilii* have fed them frozen spinach (organic), tetraselmis fish flakes (*Tetra*, 7101), and live phytoplankton [32]. We found traditional methods to be ambiguous in terms of how much spinach the worms were actually receiving (e.g. "5 gr frozen spinach" could contain variable amounts of actual spinach, depending on the frozen water content of the product). Also studying growth and other physiological processes require standardized feeding conditions that can be replicated across different labs.

We therefore set out to develop an easy-to-replicate method of feeding (Fig 7): in essence, cultures were given powdered spirulina (1.0 g/L) (*Micro Ingredients*) and Sera micron flakes (0.3 g/L) (*Sera*, 0072041678) suspended in 0.22 μm NFSW (see S1 File for recipes and volumes of food used per box size). This way the worms received a homogenous mixture of food, the volume of which could be easily adjusted if fouling was observed or if food was consumed too quickly (S3 Fig). Small, low density boxes (30 worms) were fed with 20 mL of this mixture and the larger, high density boxes (>100 worms) received 40 mL. Worms were fed twice per week, on Tuesdays and Fridays (note that labs using the spinach-tetramin regimen typically feed their cultures on a Monday-Wednesday-Friday schedule).

### Comparison of spinach and spirulina feeding regimens in juveniles

To compare the new spirulina-sera micron cocktail feeding method to the traditional spinach-fish flakes-algae feeding regimen, we tested the growth rate difference between groups of juvenile worms that were on either of these diets. For simplicity we refer to these as spinach versus

spirulina feeding regimens, even though the spinach regimen also includes fish flakes and algae, and the spirulina regimen includes sera micron. This experiment was carried out at Florian Raible's laboratory at MFPL (Vienna), where the primary feeding regimen is the spinach regimen.

For the experiment, 80 sibling juveniles (strain PIN619512 R-mix) that were 53 days old were split into eight culture boxes (500 mL, 1:1 AFSW:NFSW). Half of the boxes were fed with spinach regimen, and the other half with spirulina regimen. The spinach-fed animals received 0.5 grams of organic spinach leaves every Tuesday, and 10 mL of algae cocktail (containing 0.25 g/L finely ground Tetramin flakes in lab-cultured *Tetraselmis marina* algae solution) every Friday. These values are based on estimated averages used by the Raible Lab, since a quantifiable feeding regimen has not been established. The spirulina-fed animals received 10 mL of spirulina cocktail every Tuesday and Friday (same spirulina regimen recipe as reported above and in Fig 7).

At the time of setting up the experiment (t = 0) worms were anesthetized in 1:1 NSFW and 7.5% Magnesium Chloride ($MgCl_2$) [12] and the number of segments were counted for each individual (average number of 19.2 segments per animal, Fig 8). After this, the number of segments in all individuals were counted once every two weeks over the course of six weeks. We found that the animals that were on the spinach regimen grew new segments at a notably faster rate (6 segments per week) than animals that were on the spirulina regimen (3.6 segments per week) (Fig 8).

It is worth noting that the spinach-fed animals did not finish eating all the spinach provided each week and were therefore fed *ad libitum*, as opposed to the spirulina-fed animals that seemed to consume the algae provided in a relatively short time. This was observed by

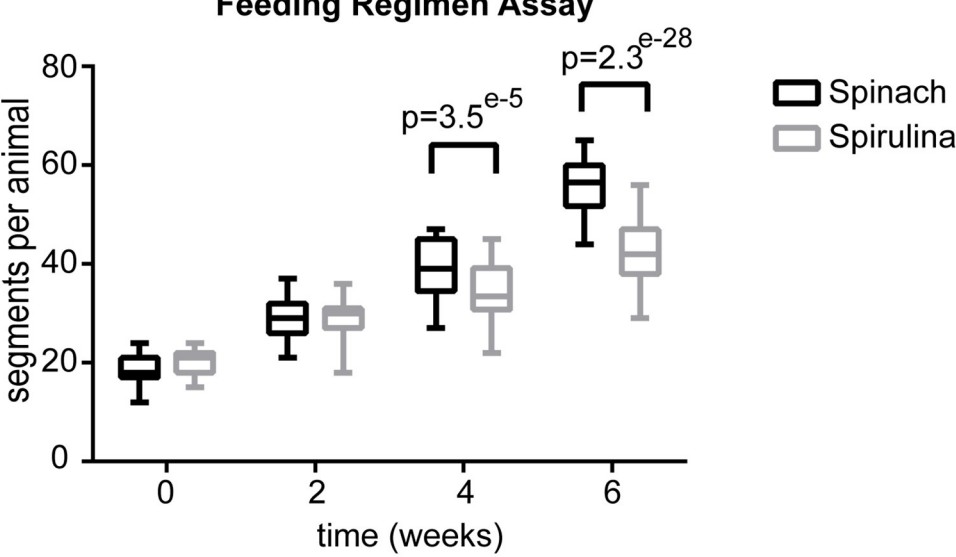

**Fig 8. Feeding regimen comparison.** Box-plot shows comparison of segment numbers in spirulina-fed and spinach-fed animals over the course of 6 weeks. To count the segments, each pair of parapodia was considered one segment. On the posterior end of the worms, only segments carrying parapodia with visible chaetae were counted. Due to natural inter-individual variation in size of the animals, the average number of segments per animal differed slightly already at t = 0, with an average of 19,2 segments per animal (SD = 2,9). Over the course of the experiment, the spinach-fed animals showed a notably faster growth rate, adding an average of 6 segments per week per animal, compared to an average of 3.6 segments per week per animal in the spirulina-fed condition. A statistically significant difference in segment numbers was observed after 4 weeks (t-test, p = $3.5e^{-5}$) and 6 weeks (t-test, p = $2.3e^{-28}$) between animals that were under spinach vs spirulina regimens.

checking the color of the water (as in S3A Fig), which turned green upon feeding and cleared up again in the first 1–2 days after feeding. This indicates that the amount of food provided to these animals was not sufficient to grant optimal growth, which may explain the slower growth rate. In a future experiment varying amounts of spirulina cocktail will be tested and compared to the spinach regimen. In the culture boxes at the MBL (Woods Hole), we have extensive algal growth over the course of only a few weeks with no adverse effects (see Fig 1G as an example). We suspect this algal film may act as an extra source of food for juvenile worms in addition to the spirulina cocktail, while during the above experiment (performed in Vienna cultures) the culture boxes did not develop such dense algal growth, which may have also affected the rate of growth. The source of this algal growth is likely due to differences in seawater sources. Also, juvenile worms in Vienna cultures are kept in 1:1 artificial and natural sea water. This potentially dilutes the amount of algae introduced into each culture box. Overall, we find that the standardized spirulina-sera micron feeding regimen provides an easily-scalable and more accurately measured feeding method, which eventually yields health mature worms (see below).

## Feeding the young larvae with spirulina

After establishing that a defined spirulina diet can be used to maintain a culture of juvenile to mature worms, we next wondered whether a similar strategy could already be applied at earlier stages. Throughout early development, *P. dumerilii* larvae depend on yolk and four large lipid droplets, one contained in each macromere as their initial source of food [17]. These droplets are largely expended by the 3-segmented juvenile stage (around 5–7 dpf), and the developing worms must seek other energy sources in order to survive. Typically, in lab cultures feeding the larvae with live algae starts around 7–10 dpf. To our knowledge, several different algae species have been used successfully by different labs. Among these are *Tetraselmis marina*, *Tetraselmis sp.* (SAG no 3.98 from Department Experimental Phycology and Culture Collection of Algae (EPSAG)), *Isochrysis galbana* (also called *T-iso*), *and Tetraselmis chuii*. Ernest E. Just procured algae for his larval *P. dumerilii* cultures by scraping the bottoms of mariculture water tables for a "felt-like growth of diatoms and protozoa" [28].

Despite being a good nutritional source, live algae cultures are prone to contamination by protozoa or rotifers, and some of these organisms can end up over-populating *P. dumerilii* larvae cultures, causing poor culturing quality. In addition, keeping live algae cultures is time-consuming. We therefore tested if a spirulina-only cocktail (powdered spirulina in NFSW, 1.0 g/L) would provide an adequate substitute for the established algae cultures for feeding larvae.

Larvae from a single batch were split by volume into 3 groups around 24 hpf. At 7 dpf, they were transferred to culture boxes with 250 mL NFSW. Each group was fed with either: i) 10 mL spirulina-only ii) 25 mL spirulina-only iii) algae-only (10 mL live algae per box), 3 times per week (Mo, Wed, Fri). At 24 days, the animals were checked under the microscope for growth and development. Larvae in both spirulina-only boxes appeared healthy. The algae-fed culture box appeared very clean at this time, suggesting the amount of algae used may not have been enough (completely consumed). These cultures were not followed for the long-term effects of the feeding regime on maturation. However, the observation suggests it may be possible to grow larvae without requiring live algae cultures, while further and long-term testing of the spirulina-only regimen is still needed.

## Worm maturation under new culturing conditions

To ensure the proper functioning of our setup, we recorded the number of maturing worms over a seven-month period (Fig 9). We did not explicitly look for mature animals outside of

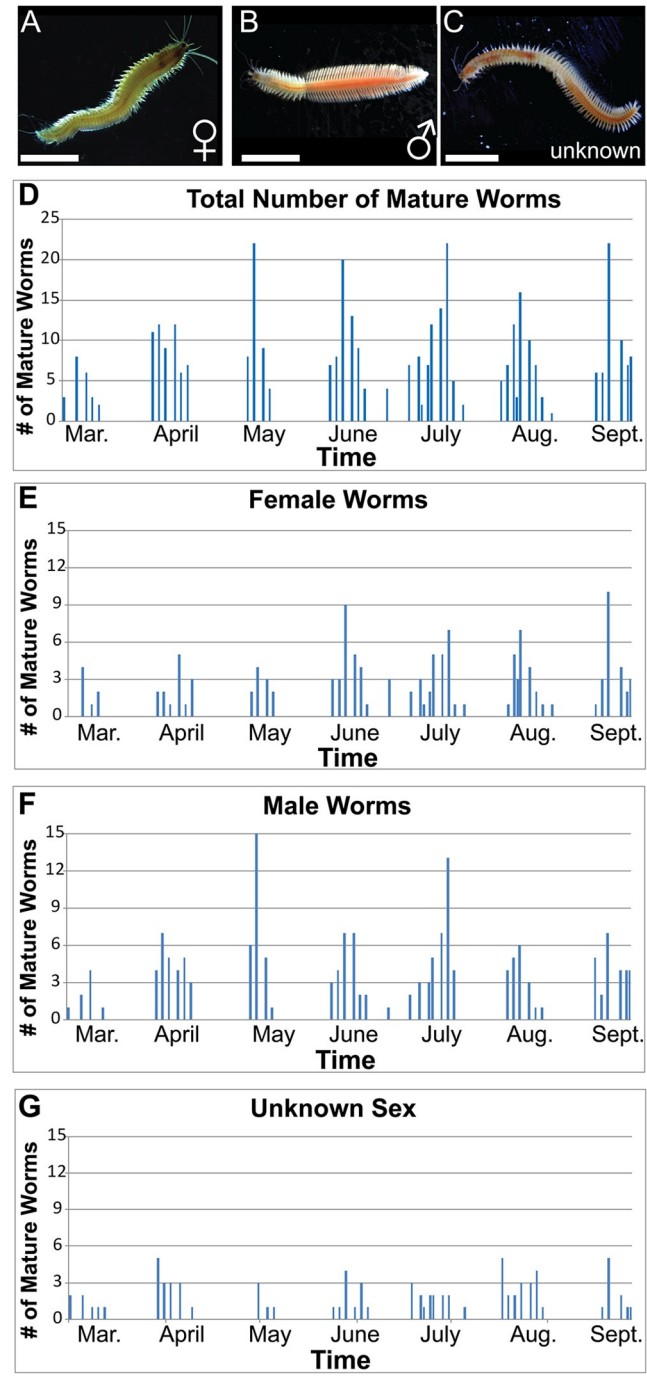

**Fig 9. *P. dumerilii* maturation.** A) Mature female *P. dumerilii*. B) Mature male *P. dumerilii*. C) Maturing *P. dumerilii* not yet categorized at the time of collection ("unknown sex"). D) Total number of mature *P. dumerilii* found for each given cycle. Culture boxes were checked 3 times per week during the maturation period. E) Number of female worms found during each cycle. F) Number of male worms found during each cycle. G) Number of worms of unknown sex at the time of collection. Note that cultures were systematically checked for mature worms only during the weeks maturation happens at a higher rate (for 2 weeks). We did not systematically check for mature worms outside of these 2 weeks. However, mature worms we came across outside of maturation weeks were noted as "off cycle" and are shown in the graphs. Scale bars: 4 mm.

the scheduled maturation window. However, off-cycle maturations were recorded if we noticed mature animals while performing other husbandry tasks. Each cycle yielded an average of 53 mature worms (Fig 9D), though the number of mature animals we found varied according to the number of worms which were in low density cultures (S4 Fig). Our cultures began producing mature animals approximately five months after we received our initial *P. dumerilii* larvae (however, this corresponds to four months after light cycles started). The initial cultures were not introduced to the lighting regimen until approximately one-month post-fertilization (December 29, 2017) when the first low density boxes were made and transferred onto the shelving unit. Prior to this, initial high density cultures had been kept in an 18C incubator in dark, without any day/night light cycle. It should therefore be noted that, typically, it is possible for mature animals to be procured more quickly (3–4 months post fertilization). At this time, we had approximately 840 worms in low density cultures (28 small boxes) and found 22 mature worms over the course of the maturation cycle. As we increased the number of low-density boxes in the shelving unit, the number of mature worms we found in the following months increased as expected. Three months after finding the first mature worms, we expanded our low-density cultures to house approximately 1500 worms, and 65 mature worms were found (Fig 9D, S4 Fig).

Our cultures tended to produce a slightly greater number of mature males (average of 24/cycle) than females (average of 18/cycle) (Fig 9E and 9F). We are uncertain if this is a product of our culturing conditions or if this mimics a more natural sex ratio for this species. Just observed the opposite in his laboratory cultures of *P. megalops*, where females outnumbered males, meanwhile he noted that in nature the reverse was the case (males outnumbered females) [28,34]. In addition, the peak maturation in our Woods Hole cultures starting at day 10 after the moon light is off differs slightly from what has been observed in some *P. dumerilii* strains in Vienna [16] and from the earliest reports from Ranzi on the Naples population [37,38]. This could be due to the method of scoring worms as mature: for example, here we used color change after metamorphosis and scored these worms as mature, while another method of scoring is by counting only the worms that have spawned as mature. Other possible reasons for the difference could be the light conditions (availability or lack of particular wavelengths), or genetics. Mature female worms were an average of 1.30 cm with a range of 0.80–2.10 cm, while males were an average of 1.44 cm with a range of 0.95–2.15 cm (S4B Fig).

## Normal development under the reported culturing conditions

Under the culturing conditions we have reported here, we have been able to raise *P. dumerilii* for several generations at the MBL, and obtained normally-developing embryos, larvae and juveniles (Fig 10). We have also injected fertilized eggs with mRNA and successfully reproduced the results obtained with past *P. dumerilii* cultures [5] (Fig 10F and 10F'). Furthermore, we have observed juvenile worms to regenerate successfully at a similar rate to the published stages of regeneration for *P. dumerilii* (results not shown) [6].

## Future directions

Future improvements to culturing *Platynereis* and other aquatic organisms remain. Integrating temperature control to the shelving setup itself (for example, by using a cooling device on each shelf) instead of controlling the temperature of the culture room will allow greater flexibility and reliability. Empirical evidence from different labs that culture *P. dumerilii* suggest that it is critical to use natural sea water for culturing embryonic and larval stages of *P. dumerilii*, while a systematic analysis has not been published to our knowledge. Other groups have observed developmental differences under artificial seawater culture conditions in other marine

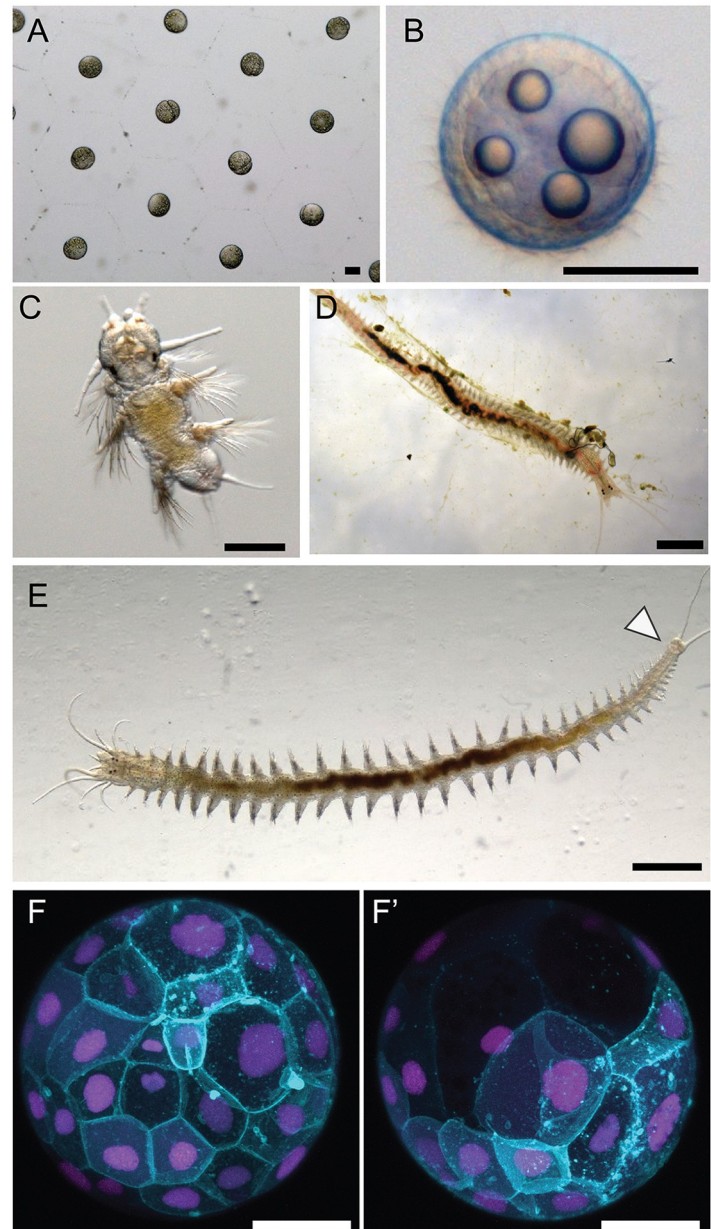

**Fig 10. *P.dumerilii* development.** A) *P. dumerilii* embryos at 2 hpf (hours post fertilization). Following fertilization, embryos secrete a protective jelly which appears as a hexagonal matrix. B) At 17 hpf *P. dumerilii* larvae reach the the prototroch stage. The larvae develop a band of cilia by 12 hpf which allows them to start swimming. C) *P. dumerilii* larva at 9 days post fertilization. D) Juveniles burrow in tubes made from a matrix of secreted mucus (Daly, 1973). E) *P. dumerilii* juveniles continue adding body segments throughout their lives from a posterior growth zone (arrowhead). F-F') Embryos injected with EGFP-caax (cyan) and Histone-mCherry (magenta) mRNA were imaged around 7 hpf using a confocal microscope. Scale bars: 100 μm (A-C), 5 mm (D,E), 50 μm (F-F').

invertebrates, with results varying by species [39]. Determining why natural sea water is needed for culturing *Platynereis* embryos and larvae, and identifying the organic and inorganic factors that make the natural sea water critical for optimal culturing outcomes may allow developing a working artificial seawater recipe and switching to artificial sea water entirely. Likewise, our monitoring currently does not include chemical compounds like nitrite or

ammonia that might accumulate in conditions of overfeeding. Determining upper bounds for such compounds might be part of a more elaborate water control system. Finally, determining whether there are more specific light spectral needs to improve *Platynereis* maturation rates will be helpful to have faster maturation times.

The benefits of the system we developed are not only scalability, but also the possibility to control food input more accurately, and the possibility to vary environmental parameters more flexibly. These are crucial components of culturing especially, as the living systems and their biology is directly affected by nutrition, and often by light cycles. Being able to control these parameters and standardize them will lead to easier reproducibility of experiments and techniques by different labs. We also envision that this culture setup design, detailed blueprints, and standardized feeding methods presented here will be beneficial to not only the community of labs that use *P. dumerilii* as a research organism, but also those labs that are interested in having *P. dumerilii* at a small scale for specific experiments and/or educational purposes, as well as for labs studying other aquatic invertebrates.

## Supporting information

**S1 Fig. Water filtration.** A) Schematic of seawater filtering process and what each level of filtration is used for. B) The 0.22 μm filtration set up. This system is more economical than plastic filter bottles. See supplementary spreadsheet for catalog numbers and ordering information.
(TIF)

**S2 Fig. Temperature.** Cultures were kept at 20˚C with some variability in the spring and summer months due to fluctuations in the temperature of the building. A portable air conditioning unit was used to help better regulate temperature during the summer months after the peaking of temperature above 22˚C in April and May.
(TIF)

**S3 Fig. Spirulina gradient.** To test which concentrations of spirulina cocktail can be fed safely to a box of 10 animals without rotting and spoiling the water, four boxes of 10 animals each were fed the following amounts of spirulina cocktail: 10, 25, 50 and 100 mL. Images were taken directly after feeding, 24 hours later and 48 hours later.
(TIF)

**S4 Fig. Number of worms in low density boxes.** A) The graph shows the number of worms living in low density cultures (30 worms/small Sterilite box) (for the period December 2017-August 2018). The number of mature animals found each cycle increased (Fig 9D) as more low density cultures were established. These numbers (along with mature animal numbers in Fig 9) can be used as a guide to scale up or down low density culture boxes, for obtaining mature animal numbers desired. B) Mature worm size box plots from 33 females and 22 males. Worms were anesthetized in a 1:1 MgCl2 –NFSW solution for approximately 5 minutes, or until swimming stopped. They were then measured under a dissecting microscope to the nearest hundredth of a centimeter.
(TIF)

**S1 File. Parts and reagents list.** This spreadsheet has the complete information on all the parts and reagents reported in this paper and their ordering information including catalog numbers and links.
(XLSX)

**S2 File. Timer setup instructions.** This file explains (with pictures) how to set up the timer that controls the light cycles.
(DOCX)

## Acknowledgments

We thank John Carr, Kate Dever, Lindsey DeMelo and the rest of the MRC staff at the MBL for their help in building the shelving setup and culture husbandry; we thank colleagues working on *P. dumerilii* for their helpful suggestions; and Guillaume Balavoine for providing the *P. dumerilii* larvae to start our cultures. We also thank two reviewers and the editor for their helpful suggestions that improved this manuscript.

## Author Contributions

**Conceptualization:** B. Duygu Özpolat.

**Data curation:** Emily Kuehn, Alexander W. Stockinger, B. Duygu Özpolat.

**Formal analysis:** Emily Kuehn, Alexander W. Stockinger, B. Duygu Özpolat.

**Funding acquisition:** Florian Raible, B. Duygu Özpolat.

**Investigation:** Emily Kuehn, Alexander W. Stockinger, B. Duygu Özpolat.

**Methodology:** Jerome Girard, B. Duygu Özpolat.

**Project administration:** B. Duygu Özpolat.

**Resources:** Jerome Girard.

**Supervision:** Florian Raible, B. Duygu Özpolat.

**Validation:** B. Duygu Özpolat.

**Visualization:** Emily Kuehn, Alexander W. Stockinger, B. Duygu Özpolat.

**Writing – original draft:** Emily Kuehn, Alexander W. Stockinger, B. Duygu Özpolat.

**Writing – review & editing:** Florian Raible, B. Duygu Özpolat.

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
