## [Decision Letter · Decision Letter 0]

27 Sep 2019

PONE-D-19-22269

A scalable culturing system for the marine annelid Platynereis dumerilii.

PLOS ONE

Dear Dr. Ozpolat,

Thank you for submitting your manuscript to PLOS ONE. After careful consideration, we feel that it has merit but does not fully meet PLOS ONE’s publication criteria as it currently stands. Therefore, we invite you to submit a revised version of the manuscript that addresses the points raised during the review process.

Both reviewers considered this work valuable and containing information useful to the research community. Both reviewers (as well as myself) have provided comments for improving the clarity of the manuscript, and I hope the authors would follow those suggestions to revise the manuscript. 

We would appreciate receiving your revised manuscript by Nov 11 2019 11:59PM. To enhance the reproducibility of your results, we recommend that if applicable you deposit your laboratory protocols in protocols.io, where a protocol can be assigned its own identifier (DOI) such that it can be cited independently in the future. For instructions see: http://journals.plos.org/plosone/s/submission-guidelines#loc-laboratory-protocols

We look forward to receiving your revised manuscript.

Kind regards,

Jr-Kai Sky Yu, Ph.D.

Academic Editor

PLOS ONE

Journal Requirements:

1. We note that Figure(s) [7] in your submission contain copyrighted images. All PLOS content is published under the Creative Commons Attribution License (CC BY 4.0), which means that the manuscript, images, and Supporting Information files will be freely available online, and any third party is permitted to access, download, copy, distribute, and use these materials in any way, even commercially, with proper attribution. For more information, see our copyright guidelines: http://journals.plos.org/plosone/s/licenses-and-copyright.

1.    You may seek permission from the original copyright holder of Figure(s) [7] to publish the content specifically under the CC BY 4.0 license.

Additional Editor Comments (if provided):

Additional comments from editor:

Line 194-195: What kind of algal culture medium was used? It would be useful to include the recipe or citation of the medium.

Line 198: Need to provide a better specification about the desired density of the algal cultures, for example, cell numbers/ml.

Line 203: How long can this algae stocks last at this refrigerated condition?

Line 209-213: How often do you conduct this check-up?

Figure 9 and its legend text, line 704- 714: I had a hard time understanding this figure; I cannot find the “black and white bars” from D-G. In addition, would it be possible to further calculate the percentage of total mature worms found in each month?

Line 723-724: It would be helpful to indicate the color (cyan and magenta) for respective mRNA injection results. Confocal “microscope”.

Reviewers' comments:

Reviewer's Responses to Questions

**Comments to the Author**

1. Is the manuscript technically sound, and do the data support the conclusions?

Reviewer #1: Yes

Reviewer #2: Yes

2. Has the statistical analysis been performed appropriately and rigorously? 

Reviewer #1: Yes

Reviewer #2: Yes

3. Have the authors made all data underlying the findings in their manuscript fully available?

Reviewer #1: Yes

Reviewer #2: Yes

4. Is the manuscript presented in an intelligible fashion and written in standard English?

Reviewer #1: Yes

Reviewer #2: Yes

5. Review Comments to the Author

Reviewer #1: The manuscript “A scalable culturing system for the marine annelid Platynereis dumerilii’ by Kuehn et al. describes in detail the set up of small culturing units for this versatile annelid system. This enables any research group with access to natural seawater to rear these animals through the entire life cycle, and to control critical parameters like day/night cycles, moon cycles, light intensities, and test various feeding regiments. Thus, this set up may also open up this animal to new areas of research including various physiological and nutritional studies.

The description of the culturing method is detailed, and mostly sufficient, and the critical areas are pointed out. Although the described culturing method reaches a higher level of flexibility and control for several parameters, the authors point out where more control could/should be achieved in the future. Thus, this set up also offers a good entry point for critical improvements that can not be tackled in current larger, less flexible systems. I recommend potential publication of the manuscript after a few concerns have been addressed.

1. One parameter that is not sufficiently defined is the seawater described in the Methods as ‘full strength natural sea water’ (lane 140). Some of the experiments were performed in a laboratory in Vienna with apparently somewhat different ‘natural sea water of lower salinity’ which the authors acknowledge in lane 501, and it is important to include better definitions for the seawater here.

2. The authors also mention the important observation that the jelly production by fertilized eggs at the MBL culture to be much longer compared to eggs produced by the ‘European’ cultures (lane 502). As this narrows the window for injections of zygotes significantly, it should be discussed that this might be a critical issue if the main purpose of an investigation of this annelid includes injection of zygotes. It would be important to include data that documents the extension of jelly production which seems quite dramatic (from 1h to 1.5h).

3. ‘unknown gender worms’ (lane 224 and Figure 9): Not sure why the authors created this category as these unknown worms will mature into females and males a little later, and could be added to these gender categories accordingly? Seems unnecessary or requires better explanation.

4. Figure 4: Black out curtain construction: As light contamination is a critical issue, I noticed that there are no Velcro strips along the bottom of the curtain (Fig 4A). Are these curtains so fitting that no light can penetrate from the bottom? How reliable is the fitting of the curtains under daily operations?

5. Figure S4: Maybe this supplemental figure would be more useful by adding also the number of maturing worms or successful fertilizations each month to the figure?

Lane 118: .. that are affected

Lane 119: rephrase sentence

Reviewer #2: General comments: This is an all-round very detailed and in-depth resource for the culturing of the important model organism Platynereis dumerilii. The paper provides a novel design for small-scale cultures that can be varied in terms of environmental conditions etc.

Overall, this manuscript provides a lot of data to support the successful establishment of a small-scale culturing system which is able to produce mature individuals in good numbers. I believe it to be an excellent starting place for those looking to establish cultures of their own at whatever scale and it provides important data to support future efforts for standardisation of culturing techniques across laboratories working with this species.

Minor comments:

1) I feel that the standardisation of feeding method by using the spirulina diet shows promise however the authors should make it clear that this method is not yet fully optimised. The results presented do not provide enough information to give a comparison between this and traditional feeding methods due to feed volumes which may or may not contribute to inferior growth rates and the different sites of the experiments. The animals on the spirulina diet were not fed ad libitum.

2) It would be useful if the authors mentioned the potential impact of several ‘unseen’ water quality parameters such as pH, nitrites, nitrates, ammonia; all of which would impact the long term health and prosperity of the culture. These parameters will potentially all be influenced by the feeding regime.

3) Do the authors also have data about the size of the worms at maturity? It would be useful to add this to the paper, if such data exist.

100 "transgenesis and genetic tools" - it would be good to mention that homozygous knockout lines have been established, first with ZFN, then TALEN, and more recently with CRISPR, and cite the relevant papers

101 "behavioral tracking [8,16]" - under behavioral tracking, it would be good to mention work also on the larval stages

101 "and live imaging [5] - please cite also work on neuronal activity imaging

118 “This is particularly important for studying biological processes are affected dramatically by nutrition” incomplete sentence

177-185: What is the size / what are the dimensions of a large box? And what volume of water is added?

182-184: What bubbling rate was used as this could impact water quality or organism behaviour. If this was not measured directly, was it gentle / vigorous etc?

223-231: Please define ‘mature worms’ here

288-290: You mention that you ‘stack’ boxes on a shelf – do you have any information as to whether this impacts culture conditions, for example light levels for the top box in the stack vs the bottom?

325: Please reference who the ‘others’ are that you followed a standardised protocol from.

423-427: Do you have any additional information or opinions on why the algal growth may have been different when comparing the two locations? For example, different sources of seawater / different filtration etc?

455-456: Do you have any data or quantification to reinforce the ‘appearance’ of a larger size of larvae?

466-467: In how many boxes / what culture size are this number of mature worms being produced?

470-473: What lighting regimen were the initial boxes on if they were not on the one you have detailed?

474-475: For these 22 mature worms, what timescale is this over? A day? A cycle?

501-503: The authors mention differences in salinity but what salinities are these and how large is the difference between the two?

511-514: I feel this needs to be discussed in a little more detail as the authors haven’t mentioned the inability to culture in artificial sea water before this point. Is there any data / information / references which compare natural and artificial sea water or discuss not being able to culture Platynereis in artificial sea water?

6. PLOS authors have the option to publish the peer review history of their article (what does this mean?). If published, this will include your full peer review and any attached files.

Reviewer #1: No

Reviewer #2: No

---

## [Author Response · Author response to Decision Letter 0]

10 Nov 2019

Please see the Response to Reviewers file attached along with the other documents.

---

## [Decision Letter · Decision Letter 1]

21 Nov 2019

A scalable culturing system for the marine annelid Platynereis dumerilii.

PONE-D-19-22269R1

Dear Dr. Ozpolat,

We are pleased to inform you that your manuscript has been judged scientifically suitable for publication and will be formally accepted for publication once it complies with all outstanding technical requirements.

With kind regards,

Jr-Kai Sky Yu, Ph.D.

Academic Editor

PLOS ONE

Additional Editor Comments (optional):

Reviewers' comments:

Reviewer's Responses to Questions

**Comments to the Author**

1. If the authors have adequately addressed your comments raised in a previous round of review and you feel that this manuscript is now acceptable for publication, you may indicate that here to bypass the “Comments to the Author” section, enter your conflict of interest statement in the “Confidential to Editor” section, and submit your "Accept" recommendation.

Reviewer #1: All comments have been addressed

2. Is the manuscript technically sound, and do the data support the conclusions?

Reviewer #1: Yes

3. Has the statistical analysis been performed appropriately and rigorously? 

Reviewer #1: Yes

4. Have the authors made all data underlying the findings in their manuscript fully available?

Reviewer #1: Yes

5. Is the manuscript presented in an intelligible fashion and written in standard English?

Reviewer #1: Yes

6. Review Comments to the Author

Reviewer #1: Kuehn et al. address every of my previous minor concerns in the revised manuscript “A scalable culturing system for the marine annelid Platynereis dumerilii’ providing a detailed valuable description of a set up for small culturing units for this versatile annelid system. For all the reasons that I mention in my previous review I recommend publication of the manuscript in PloS ONE.

7. PLOS authors have the option to publish the peer review history of their article (what does this mean?). If published, this will include your full peer review and any attached files.

Reviewer #1: No

---

## [Editor Report · Acceptance letter]

26 Nov 2019

PONE-D-19-22269R1 

A scalable culturing system for the marine annelid *Platynereis dumerilii*. 

Dear Dr. Ozpolat:

I am pleased to inform you that your manuscript has been deemed suitable for publication in PLOS ONE. Congratulations! Your manuscript is now with our production department. 

With kind regards,

on behalf of

Dr. Jr-Kai Sky Yu 

Academic Editor

PLOS ONE